# Efficacy, Immunogenicity, and Safety of Pertussis Vaccine During Pregnancy: A Meta-Analysis

**DOI:** 10.3390/vaccines13070666

**Published:** 2025-06-20

**Authors:** Qianqian Shi, Jun Li, Quanman Hu, Cheng Cheng, Kun Yang, Xiaoyu Li, Xiaoru Song, Shuaiyin Chen, Guangcai Duan

**Affiliations:** 1Department of Epidemiology and Statistics, College of Public Health, Zhengzhou University, Zhengzhou 450001, China; qqshi@gs.zzu.edu.cn (Q.S.); quanmanhu@gs.zzu.edu.cn (Q.H.); kunyang123@gs.zzu.edu.cn (K.Y.); xyli@gs.zzu.edu.cn (X.L.); xrsong@gs.zzu.edu.cn (X.S.); gcduan@zzu.edu.cn (G.D.); 2Henan Center for Disease Control and Prevention, Zhengzhou 450016, China; henan_epi2@163.com; 3Henan Cancer Hospital, Zhengzhou 450008, China; chengc2016zzu@126.com

**Keywords:** pregnancy, infants, pertussis

## Abstract

Background: A growing number of countries implement prenatal pertussis vaccination policies to safeguard unvaccinated infants. This meta-analysis aimed to evaluate the efficacy, immunogenicity, and safety of antenatal Tdap vaccination in pregnant individuals. Methods: We systematically searched PubMed, Embase, and Web of Science from their inception to 16 February 2025, rigorously screening studies and including seven randomized controlled trials and 10 case-control studies published between 2014 and 2024. For the test-negative design meta-analysis, odds ratios with 95% confidence intervals served as effect estimates, and vaccine efficacy was calculated accordingly. Standardized mean differences were used to assess geometric mean concentrations, while relative risks evaluated safety. Results: Maternal vaccination during pregnancy demonstrated 85% vaccine effectiveness (95% CI: 78–89%) in protecting infants under 3 months old. Pooled standardized mean differences for cord blood IgG antibodies against pertussis toxin, pertactin, and filamentous hemagglutinin were 1.57 (95% CI: 1.25–1.89), 2.15 (95% CI: 1.82–2.48), and 2.25 (95% CI: 1.81–2.68), respectively, indicating higher antibody levels in infants of vaccinated women before their first immunization. Safety analysis showed no significant association between Tdap vaccination during pregnancy and serious adverse events in infants (RR = 0.76, 95% CI: 0.46–1.24) and pregnant women (RR = 1.22, 95% CI: 0.83–1.81). Conclusion: Our findings support the implementation of pertussis vaccination during pregnancy.

## 1. Introduction

Pertussis, an acute respiratory infection caused by *Bordetella pertussis*, remains a global public health challenge, particularly for infants and young children. Its high infectivity makes susceptible infants vulnerable to disease once exposed, with the highest incidence in infants under 3 months of age [1,2,3]. Infection in this age group not only leads to severe illness and family burden but also underscores the urgent need for maternal immunization to prevent infant deaths [4,5]. In high-income nations, the annual pertussis-related hospitalization rates for infants under 6 months old range from 100 to 1000 cases per 100,000 population. Data from the United States and Australia indicate that more than 60% of pertussis-associated hospitalizations occur in children under one year of age [6,7,8]. Although the burden of pertussis has decreased since the introduction of whole-cell and acellular vaccines for children, morbidity and mortality remain high in young children, who account for 70–90% of all pertussis hospitalization and deaths [9,10]. As the World Health Organization (WHO) recommends that infants should be vaccinated against pertussis from 6–8 weeks of age [11], neonates remain vulnerable to the disease for the first few weeks of life. In this context, maternal pertussis vaccination (such as Tdap) during pregnancy confers passive immunity to the newborn through placental antibody transfer and becomes a key intervention to bridge the “immune gap period” [12,13,14,15].

Following the 2012–2013 pertussis epidemic, the United Kingdom and the United States recommended for the first time that pregnant women be vaccinated against pertussis in reaction to the increasing cases of pertussis in the general population and numerous infant fatalities [16,17]. In 2015, the World Health Organization (WHO) officially recommended maternal immunization using acellular vaccines against pertussis, a recommendation rooted in studies on acellular primary immunization [18]. By 2024, this practice had been endorsed in 58 countries worldwide, reflecting a growing global consensus on its efficacy in protecting vulnerable newborns [1]. Nevertheless, to comprehensively evaluate the practicality and significance of administering pertussis vaccines during pregnancy and delve into several critical aspects, the safety of the vaccine must be the primary concern. For example, whether vaccination causes serious adverse reactions in mothers and babies, and whether there is an increased risk of premature birth, miscarriage, and fetal abnormalities [19], carefully designed randomized controlled trials are needed for accurate analysis, and the use of vaccine and control groups minimizes the effect of confounding factors and provides reliable safety evidence. The study by Aidibai Simayi et al. conducted a systematic review of the safety and efficacy of the pertussis vaccine during pregnancy [20], but there was a lack of studies on the effectiveness in infants younger than 3 months. In addition, Nguyen et al. added to the analysis of morbidity in infants under 3 months of age [21], but included only a limited number of emerging studies. It is important to know the effectiveness of the vaccine in infants up to 3 months of age. In less than two years, seven new relevant studies have been published [1,22,23,24,25,26,27], highlighting the recent interest in investigating this issue.

This study first evaluates the protective efficacy of maternal pertussis vaccination against infant infection, analyzing the safety profiles, immunogenicity, and vaccine effectiveness in 3-month-old infants. By synthesizing these empirical findings, the research aims to provide a scientific basis for optimizing the prenatal vaccination strategies and informing efforts to improve maternal and infant health outcomes.

## 2. Materials and Methods

### 2.1. Search Strategy

PubMed, Embase, and Web of Science were systematically searched using medical subject headings (MeSH) and keywords. The search period extended from the databases’ initial publication dates to 16 February 2025. The search approach was built upon the following keywords: “pertussis”, “whooping cough”, “pregnancy”, “efficacy”, “safety”, “baby”, etc. The specific retrieval strategies are set out in Appendix A.

### 2.2. Inclusion and Exclusion Criteria

Studies meeting the inclusion criteria were as follows: (a) randomized controlled trials (RCTs); (b) case-control studies (CCSs); (c) experimental groups receiving Tdap vaccination during pregnancy and control groups receiving a placebo, standard vaccination, or no intervention; (d) the analysis evaluated maternal vaccination timing associations with three key outcomes: safety profiles, immune response parameters, and clinical efficacy metrics; (e) there were clear diagnostic criteria in the case group and control group. We removed the following: (a) cohort studies; (b) papers not related to the subject; (c) repeated publications; (d) research focused on animals or laboratory experiments.

### 2.3. Quality Assessment

Quality assessment was independently conducted by two reviewers following the Cochrane Collaboration guidelines. For RCTs, the risk of bias was evaluated using the Cochrane RoB 2 tool (version 2 of the risk-of-bias tool for randomized trials) [28]. The Newcastle–Ottawa Scale (NOS) was employed to assess the quality of nonrandomized studies. The NOS scoring criteria defined high quality as ≥7 points, moderate quality as 5 to <7 points, and low quality as <5 points [29]. The third author resolved all disagreements.

### 2.4. Data Extraction

A rigorous independent evaluation process was implemented to systematically retrieve pertinent information from all the qualifying studies, with the following datasets methodically obtained from each investigation: (a) first author’s name; (b) journal and country of publication; (c) type of study; (d) type of vaccination; (e) number of subjects in each group; (f) gestational age at vaccination; (g) trial registration number; (h) start and end dates of the study; (i) leading vaccine manufacturer. For the evaluation of safety, we retrieved the occurrence rate of SAEs in the pregnant and infant groups. Serious adverse events (SAEs) monitored in the pregnant women group included maternal death, gestational hypertension, severe infections, preeclampsia, postpartum hemorrhage, and preterm birth. For infants, SAEs comprised neonatal death, congenital cardiac defects, respiratory distress syndrome, severe pneumonia, hyperbilirubinemia requiring phototherapy, febrile convulsions, dehydration, and hypoglycemia necessitating medical intervention. To assess maternal pertussis immunogenicity, geometric mean concentrations of antipertussis toxin (anti-PT), antifilamentous hemagglutinin (anti-FHA), and antipertactin (anti-PRN) immunoglobulin G (IgG) antibodies were extracted from cord blood at delivery and from all vaccine antigens before the first vaccination. The vaccine efficacy was assessed in early infancy.

### 2.5. Statistical Analysis

This review was conducted in accordance with the PRISMA guidelines and was registered in the International Prospective Register of Systematic Reviews (PROSPERO, CRD420251068024). Statistical computations were performed employing STATA 17.0 (Stata Corp, College Station, TX, USA), adopting *α* = 0.05 as the significance threshold. Vaccine efficacy estimations utilized odds ratios (OR) with 95% confidence intervals (CI), calculated through the formula: VE = (1 − OR) × 100%. As most studies report geometric mean antibody levels, these were ln-transformed, and all the studies used lognormal distributions [30]. The GMC of anti-PT, anti-FHA, and anti-PRN IgG antibodies was calculated using the concentration conversion average of anti-LN, and the GMC was measured using the SMD. SAEs were evaluated using the risk ratio (RR) and 95% CI. The random effects model was used to estimate the combined effect size [31]. The *I*^2^ statistic was used to gauge heterogeneity, where an *I*^2^ value exceeding 50% or *p* < 0.05 signaled significant statistical heterogeneity. A sensitivity analysis was employed in order to evaluate the stability of the meta-analysis results. The Egger test and the Begg test were used to detect publication bias, and *p* ≤ 0.05 was taken to indicate the existence of publication bias. When publication bias was present, the shear compensation method was used to see how much the unpublished literature affects the results of a meta-analysis [32].

## 3. Results

### 3.1. Study Selection and Quality of the Studies

A total of 4608 articles were initially identified, with 17 studies ultimately meeting the inclusion criteria: six assessed safety, seven evaluated immunogenicity, and ten examined vaccine effectiveness. The screening workflow is detailed in Figure 1.

The results are presented in Table 1 and Table 2. All trials were published between 2014 and 2024. In all the trials that were assessed for safety [19,24,33,34,35,36], 499 serious adverse events during pregnancy were reported, of which 267 were from the experimental group and 232 from the control group. In all the studies evaluating immunogenicity [19,24,33,34,35,36,37], the number of women recruited into the experimental (Tdap) and control groups was 765 and 682, respectively. Of all the studies evaluating vaccine efficacy, eight studies [8,25,26,38,39,40,41,42] reported efficacy in infants aged two months, and two studies [43,44] reported efficacy in infants aged three months old.

In general, seven RCTs [19,24,33,34,35,36,37] were assessed for bias using RoB 2. Moreover, 10 case control studies [8,25,26,38,39,40,41,42,43,44] were assessed for bias using the NOS. Appendix A provide detailed summaries of each study’s quality in the meta-analysis.

### 3.2. Efficacy of Vaccines for Infants Under 3 Months of Age

We analyzed the overall VE/OR and 95% CI for infants 0–3 months of age whose mothers were vaccinated during pregnancy: the overall VE is 85% (95% CI: 78–89%), and the overall OR is 0.15 (95% CI: 0.11–0.22, *I*^2^ = 0.0%, *p* = 0.756; Figure 2).

### 3.3. Vaccine Immunogenicity

The analysis of the anti-PT and anti-PRN IgG geometric mean concentrations in the cord blood of infants incorporated seven studies. Additionally, five studies presented the relevant GMCs for the anti-FHA. The study by Perrett et al. divided the time of vaccination during pregnancy into 27–32 weeks and 33–36 weeks and reported the results separately [33], so we have included the results of the two parts of the study separately. The combined SMDs of PT, PRN, and FHA IgG from the cord blood were 1.57, 95% CI: 1.25–1.89, *I*^2^ = 84.6%, *p* < 0.001 (Figure 3A), 2.15, 95% CI: 1.82–2.48, *I*^2^ = 83.1%, *p* < 0.001 (Figure 3B), and 2.25, 95% CI: 1.81–2.68, *I*^2^ = 88.2%, *p* < 0.001 (Figure 3C), respectively. The results indicate that there were higher GMCs of antibodies against pertussis in the Tdap group in comparison with the control group. We also conducted a meta-analysis of anti-PT, anti-PRN, and anti-FHA before the first Tdap vaccination in infants. This partial result is shown in Figure 4.

Due to the high heterogeneity of the results of the immunogenicity analysis, we conducted a subgroup analysis. In the subgroup analysis stratified by vaccine manufacturers (GSK and SP), significant heterogeneity was observed in the antibody responses across the subgroups. For the anti-PT antibodies in cord blood, the pooled standardized mean difference (SMD) for the GSK subgroup (four studies) was 1.86 (95% CI: 1.56–2.17), significantly higher than the SP subgroup’s SMD of 1.23 (95% CI: 0.89–1.57), with statistically significant between-group interaction (*p* = 0.036). In the analysis of anti-FHA antibodies in cord blood, the GSK subgroup (four studies) exhibited a pooled SMD of 2.59 (95% CI: 2.27–2.91), which was significantly greater than the SP subgroup’s SMD of 1.75 (95% CI: 0.98–2.51, *p* < 0.001), with a trend toward statistical significance in interaction effect (*p* = 0.058). The intervention type of the control group had a limited impact on the heterogeneity. For the anti-PRN antibodies in cord blood, the placebo subgroup (five studies) exhibited a pooled SMD of 2.18 (95% CI: 2.02–2.34) with no heterogeneity (*I*^2^ = 0%), whereas the TT/Td subgroup (three studies) showed substantial heterogeneity (*I*^2^ = 94.5%); however, the between-group interaction was not statistically significant (*p* = 0.952). In the analysis of anti-FHA antibodies, the placebo subgroup (four studies) had a higher pooled SMD (2.49, 95% CI: 2.10–2.87) compared to the TT/Td subgroup (1.97, 95% CI: 1.12–2.82), though the interaction effect remained non-significant (*p* = 0.329). For vaccination gestational age (≥30 weeks and <30 weeks), a trend toward higher antibody levels was observed in the ≥30 weeks subgroup, though statistical significance was not reached. In the analysis of the anti-FHA antibodies in cord blood, the pooled SMD for the ≥30 weeks subgroup (five studies) was 2.40 (95% CI: 2.07–2.73), higher than the <30 weeks subgroup’s SMD of 1.88 (95% CI: 0.28–3.49, two studies), with a non-significant interaction effect (*p* = 0.367). Similarly, for anti-PT antibodies, the ≥30 weeks subgroup (six studies) exhibited a pooled SMD of 1.65 (95% CI: 1.24–2.05), greater than the <30 weeks subgroup’s SMD of 1.37 (95% CI: 0.91–1.82, two studies), but the between-group interaction remained non-significant (*p* = 0.489). The specific analysis results are included in the Appendix A.

An analysis of antibody levels in babies around one year old showed that there were significantly fewer anti-PT and anti-FHA antibodies in the Tdap group compared with the control group. No significant intergroup difference was found in anti-PRN antibody levels. This analysis result is presented in the Appendix A.

### 3.4. Vaccine Safety

#### 3.4.1. Infant SAE Analysis

Five studies [19,24,33,34,36] reported infant SAEs. The initial pooled analysis showed no significant difference between the groups (RR = 0.76, 95% CI: 0.46–1.24), but with high heterogeneity (*I*^2^ = 71.4%, *p* = 0.007; Figure 5A). Sensitivity analysis excluding Munoz FM 2014 (RR = 0.32, 95% CI: 0.15–0.67) reduced the heterogeneity to moderate levels (*I*^2^ = 42.3%, *p* = 0.158) and yielded a pooled RR of 0.95 (95% CI: 0.66–1.38), which remained non-significant. This supports the robustness of the primary conclusion despite the residual heterogeneity.

#### 3.4.2. Pregnant Women SAE Analysis

Six studies [19,24,33,34,35,36] reported SAEs in pregnant women. The meta-analysis of combined datasets revealed no statistically significant intergroup differences (RR = 1.22, 95% CI: 0.83–1.81), with low heterogeneity (*I*^2^ = 52.9%, *p* = 0.060; Figure 5B).

### 3.5. Sensitivity Analysis and Publication Bias

Sensitivity analysis demonstrated that the pooled risk ratios, standardized mean differences, and odds ratios remained consistent following the sequential exclusion of individual studies, confirming the robustness of the findings. The results of this part of the analysis are presented in the Appendix A. The potential for publication bias in the included studies was evaluated through the utilization of Begg’s test, Egger’s test, and funnel plots. Relevant funnel plots are shown in Appendix A. The *p* values for both tests were >0.05, indicating the absence of statistically significant publication bias.

## 4. Discussion

This meta-analysis synthesized data from seven RCTs and 10 CCSs, encompassing 842 pregnant women who received Tdap vaccination during gestation and 607 infant cases analyzed in CCSs. Through a systematic review and synthesis of the existing evidence on antenatal pertussis immunization, the findings provide valuable insights for optimizing prenatal vaccination protocols and strengthening dual protection for both maternal and neonatal health outcomes.

A systematic review by Weerdt et al. collated global Tdap vaccination practices during pregnancy, documenting 22 distinct administration schedules across 58 countries [1]. While the optimal gestational timing remains undetermined, this heterogeneity suggests an emerging trend toward routine antenatal pertussis immunization. By synthesizing the data from 10 case-control studies, the analysis revealed that maternal vaccination during pregnancy conferred an 85% vaccine effectiveness (VE, 95% CI: 78–89%) in preventing pertussis infection in infants under 3 months of age, a vulnerable period before primary immunization is initiated. Notably, this protective effect translated directly into clinical outcomes: an Australian case-control study demonstrated a 94% reduction in hospitalization risk (95% CI: 59–99%) among infected infants whose mothers received Tdap, highlighting the vaccine’s efficacy in preventing severe disease requiring medical intervention [44]. Concurrently, a UK study reported a 93% reduction in pertussis risk among newborns of vaccinated mothers, further linking maternal immunity to the diminished severity of clinical presentations [38]. These robust findings substantiate the significant protective effect of maternal vaccination against early infant pertussis infection, providing empirical justification for nations yet to implement antenatal pertussis immunization policies.

The immunogenicity analysis yielded positive outcomes. Geometric mean concentrations of anti-PT, anti-PRN, and anti-FHA IgG antibodies in umbilical cord blood demonstrated statistically significant pooled standardized mean differences, consistent with the findings from the included studies [19,24,33,34,35,36,37]. This confirms that maternal vaccination effectively induces immune responses and facilitates transplacental antibody transfer, conferring neonatal immune protection at birth. This finding not only demonstrates the vaccine’s efficacy in maternofetal immune transfer, but also elucidates the immunological rationale underlying antenatal vaccination as a critical preventive intervention against early-life pertussis infection in infants. Due to the observation of a large amount of heterogeneity in the study of vaccine immunogenicity, we conducted subgroup analysis. Taking umbilical cord blood anti-PT antibody as an example, the standardized mean difference of the GSK vaccine group was 1.86 (95% CI: 1.56–2.17), which was significantly higher than that of the SP group (1.23, 95% CI: 0.89–1.57, *p* = 0.036). This finding may be associated with compositional differences between GlaxoSmithKline (three components: PT, PRN, and FHA) and Sanofi Pasteur (five components: PT, PRN, FHA, DT, and TT, Fimbriae 2 and 3 combined) vaccines, where tetanus and diphtheria toxoids in the formulations might influence pertussis component immunogenicity [21,45,46]. Furthermore, the antigen purity or adjuvant formula of GSK vaccines may be superior, thereby inducing a stronger IgG antibody response [34]. A similar trend was also observed in anti-FHA antibodies (SMD = 2.59 in the GSK group vs. 1.75 in the SP group, *p* = 0.058). However, the inter-manufacturer differences in anti-PRN antibodies were not statistically significant (*p* = 0.077), which may be attributed to the smaller number of included studies (only four) or the more stable immunogenicity of PRN antigens. The impact of vaccine types used in the control group on the heterogeneity differed across the antibody types. In the analysis of anti-PRN antibodies, the heterogeneity in the placebo subgroup was extremely low (*I*^2^ = 0%), whereas severe heterogeneity was observed in the TT/Td subgroup (*I*^2^ = 94.5%). This may be attributed to the potential cross-reactivity between tetanus toxoids in TT/Td vaccines and pertussis antigens, or substantial variations in the production processes and vaccination doses of TT/Td vaccines across the studies, leading to result fluctuations [35]. However, interaction tests showed no statistically significant intergroup differences (*p* = 0.952), indicating that the intervention type in the control group was not the primary source of the heterogeneity.

After stratification by vaccination gestational age, the antibody levels in the ≥30 weeks group were generally higher than those in the <30 weeks group, although the heterogeneity remained significant. This may be associated with the higher efficiency of placental antibody transport during the third trimester of pregnancy [12]. However, the <30 weeks group had a small sample size (only two studies), limiting the stability of the results. Notably, even in the ≥30 weeks group, the heterogeneity was not completely eliminated, suggesting that factors such as individual immune status and genetic background may influence the vaccine response in addition to the gestational age [36]. Differences in research design, antibody detection methods, and follow-up duration may all contribute to the heterogeneity. For example, Perrett et al. [33] subdivided the vaccination timing into 27–32 weeks and 33–36 weeks of gestation and found higher antibody levels in the latter subgroup, further supporting the impact of gestational age on immunogenicity. However, the inclusion of this study as two separate datasets in the analysis might have amplified the statistical heterogeneity.

Moreover, vaccination-induced passivation is another critical issue that demands our attention. Our research findings indicated that, at one year of age, the anti-PRN antibody levels in infants within the Tdap group were comparable to those in the control group. Nevertheless, the anti-PT and anti-FHA antibody levels were lower than those in the control group. PT and FHA are key virulence factors of Bordetella pertussis. The observed reduction in their corresponding antibody levels suggests that the humoral immune protection in these infants may wane as they age. While cellular immunity also contributes to long-term defense, antibody titers remain a core surrogate marker for evaluating clinical protection. The diminished anti-PT and anti-FHA levels at 12 months old could potentially increase the risk of pertussis infection during the second year of life. However, definitive “protective antibody thresholds” remain undefined, necessitating further investigation into the association between specific antibody levels and the actual infection risk. Notably, the stability of anti-PRN antibodies indicates that PRN-mediated immune protection may be more sustained within the first year. Nevertheless, effective pertussis protection relies on the synergy between humoral and cellular immunity. The stability of a single antibody component does not fully represent the overall immune status. Future studies should delve into the multidimensional correlates of immune protection.

In conclusion, the dynamic changes in antibodies observed in 12-month-old infants born to Tdap-vaccinated mothers underscore the complexity of infant immune protection following maternal vaccination. These findings provide a crucial reference for refining pertussis immunization strategies and enhancing long-term prevention and control. Future validation through large-sample, long-term follow-up studies remains imperative.

In terms of vaccine safety profiles, our analysis demonstrated no statistically significant associations between prenatal Tdap vaccination and (SAEs) in either infants (RR = 0.76, 95% CI: 0.66–1.24) or pregnant women (RR = 1.22, 95% CI: 0.83–1.81), with the findings aligning with the pooled safety analyses from the included studies [19,24,33,34,35,36]. Notably, while Perrett et al. reported an isolated case of preterm birth potentially attributable to saline placebo administration in the control group, this observation did not alter the overall safety conclusions derived from the aggregated evidence [33]. Subgroup analysis identified the study by Muñoz et al. as a significant outlier in infant SAE assessments, demonstrating a substantially lower risk ratio (RR = 0.32) compared to the pooled estimate [19]. A visual inspection of the forest plot revealed marked deviation from the homogeneity axis, with subsequent exclusion of this trial resulting in a statistically significant reduction in the heterogeneity indices. This sensitivity analysis suggests that this study may be a major source of heterogeneity in the infant SAE dataset. Notwithstanding the considerable heterogeneity in the primary analyses, the sensitivity analyses confirmed the robustness of the conclusions, demonstrating that antenatal Tdap administration is not associated with statistically significant increases in serious adverse event risks in infants. The analysis of maternal SAEs demonstrated nonsignificant risk differences under low heterogeneity, reinforcing the safety of antenatal Tdap immunization. These findings alleviate the concerns regarding vaccine-associated severe adverse outcomes in pregnant populations, thereby supporting the secure implementation of maternal vaccination programs.

This study has several limitations. Despite the implementation of rigorous search strategies and predefined inclusion/exclusion criteria, the final analysis incorporated only 17 studies, predominantly from high-income countries. This geographic concentration potentially restricts the generalizability of the findings to low- and middle-income countries (LMICs), where healthcare systems, vaccination protocols, and maternal–fetal immune dynamics may differ substantially. Furthermore, the substantial heterogeneity observed across certain studies, while addressed through sensitivity analyses to ensure result robustness, necessitates further investigation into the potential sources such as regional population characteristics and the variations in vaccination protocols to enhance the interpretability.

## 5. Conclusions

The findings of this meta-analysis support the safety and efficacy of antenatal pertussis vaccination, establishing crucial immunological protection for infants too young for direct immunization. Our findings provide robust empirical support for implementing maternal pertussis vaccination programs while offering evidence-based justification for policy formulation in regions yet to adopt this preventive strategy.

## Figures and Tables

**Figure 1 vaccines-13-00666-f001:**
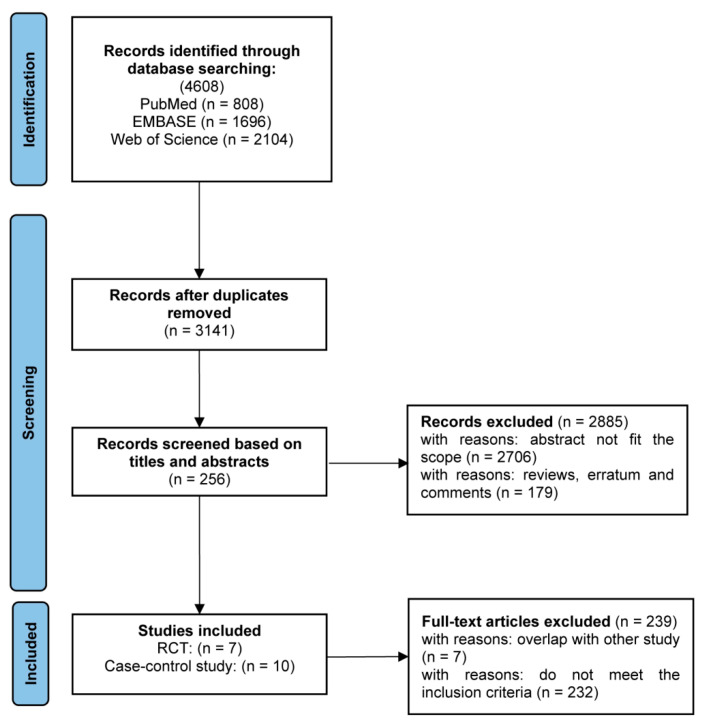
Flow chart of the literature searching process.

**Figure 2 vaccines-13-00666-f002:**
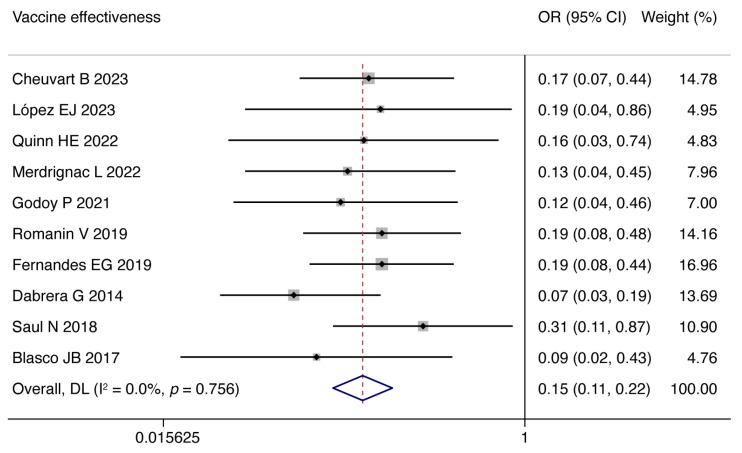
Forest plots of vaccine effectiveness [8,25,26,38,39,40,41,42,43,44].

**Figure 3 vaccines-13-00666-f003:**
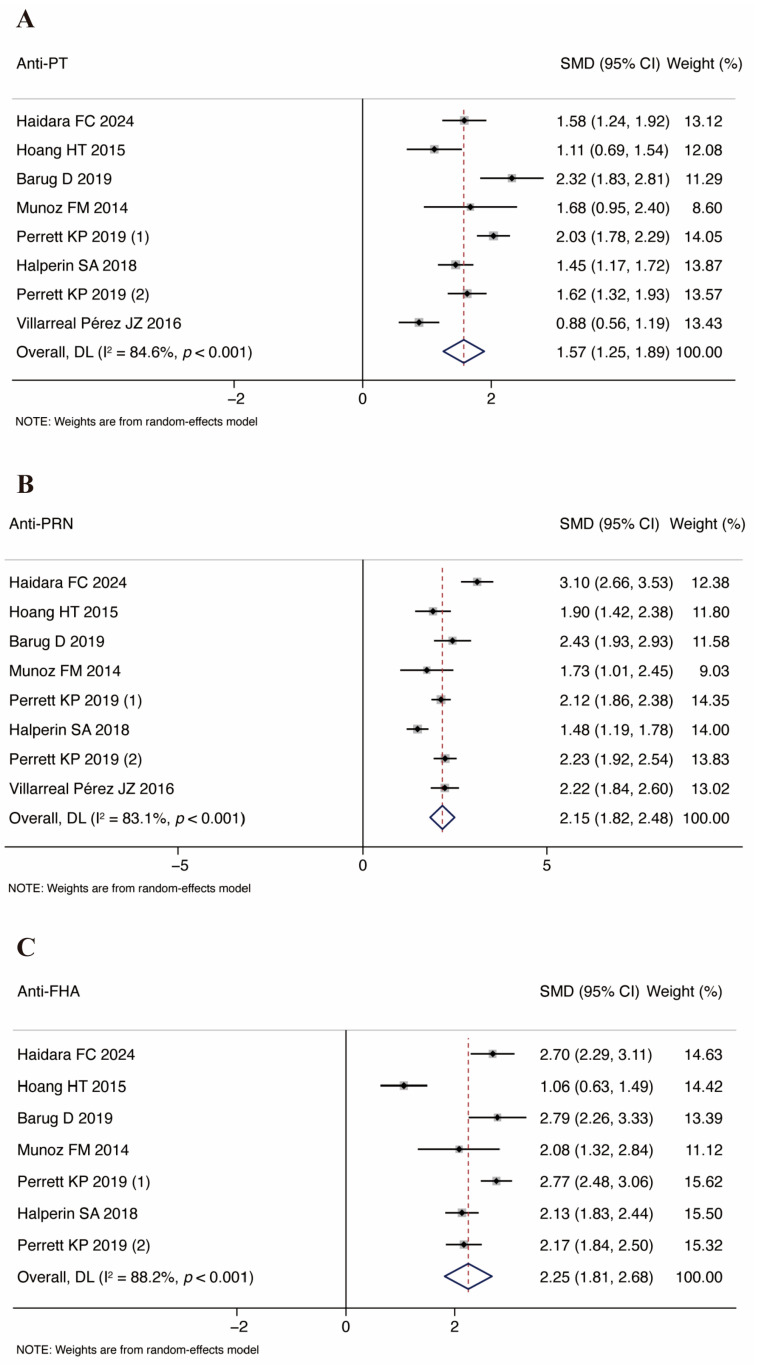
Forest plots of GMCs for pertussis antibodies in infants from cord blood. Note: (**A**) anti-PT; (**B**) anti-PRN; (**C**) anti-FHA [19,24,33,34,35,36,37].

**Figure 4 vaccines-13-00666-f004:**
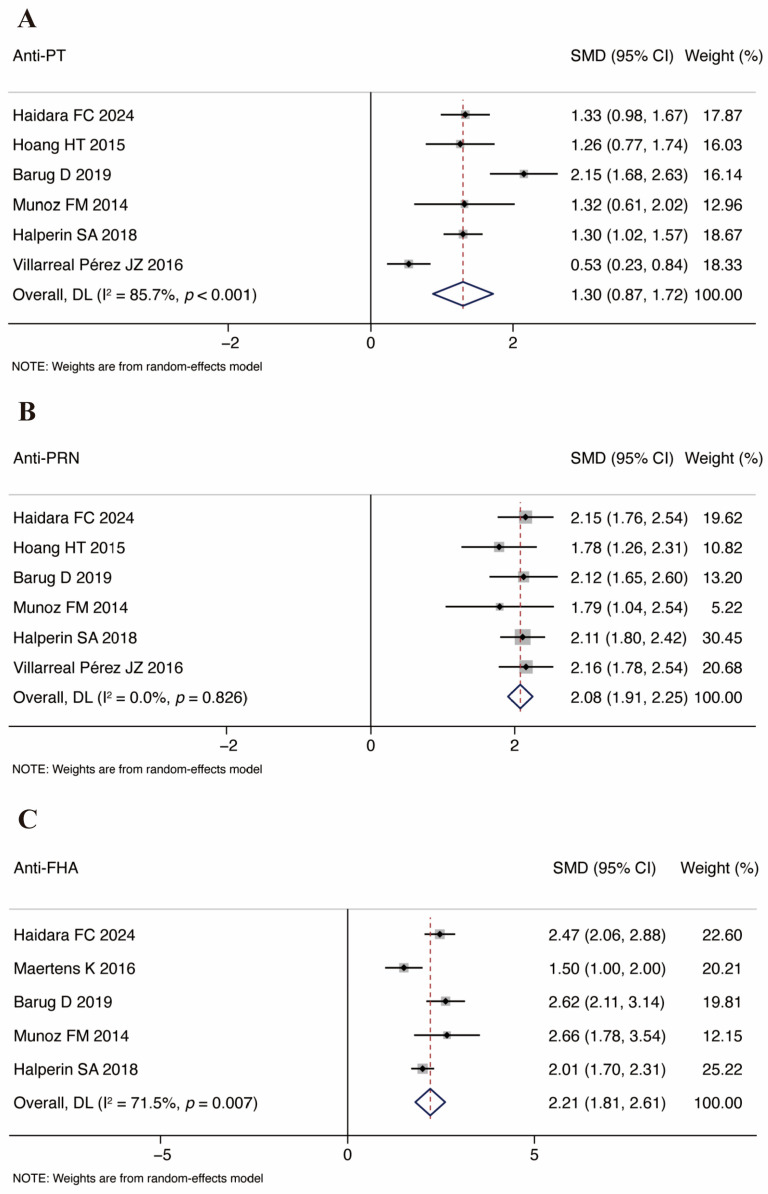
Forest plots of GMCs for pertussis antibodies before primary vaccination. Note: (**A**) anti-PT; (**B**) anti-PRN; (**C**) anti-FHA [19,24,34,35,36,37].

**Figure 5 vaccines-13-00666-f005:**
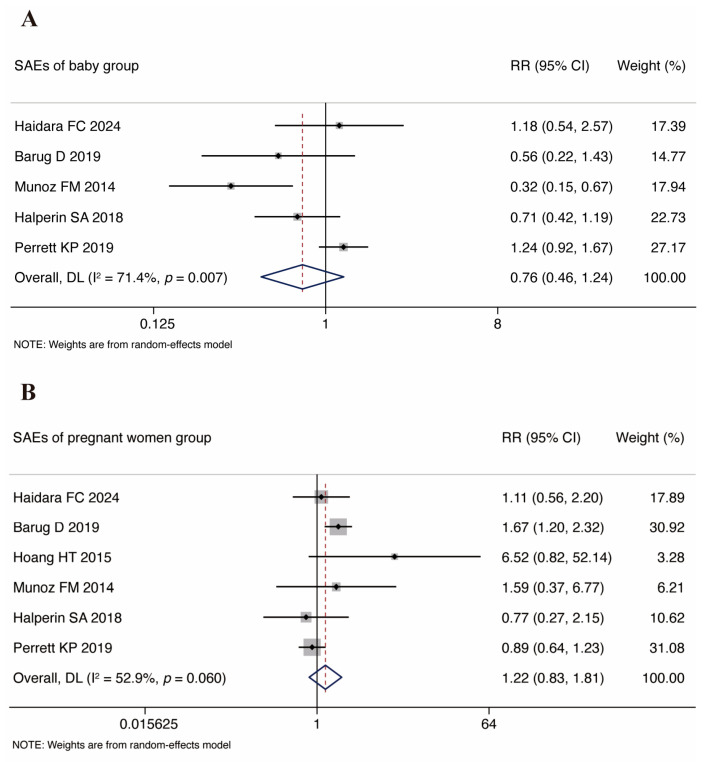
Forest plots of SAEs of women and infants. Note: (**A**) baby group; (**B**) pregnant women group [19,24,33,34,35,36].

**Table 1 vaccines-13-00666-t001:** Characteristics of included randomized controlled studies.

Study ID	Study Period	Study Design	Intervention of Women (Exposure/Control)	N of Women (Exposure/Control)	Country	Gestational Age in Weeks of Vaccination	Major Vaccine Manufacturers	Registration No.
Perrett et al. [33]	2015–2017	RCT obsever-blind	Tdap/placebo	341/346	Australia, Canada, Czech Republic, Finland, Italy, and Spain	27–36 weeks	GSK	NCT02377349
Barug et al. [34]	2014–2016	RCT open-label	Tdap/--	58/60	Netherlands	30–32 weeks	GSK	EudraCT 2012-004006-9/NTR4314
Munoz et al. [19]	2008–2012	RCT double-blind	Tdap/placebo	33/15	The United States	30–32 weeks	SP	NCT00707148
Villarreal Pérez et al. [37]	2011–2014	RCT double-blind	Tdap/placebo	90/81	Vietnam	30–32 weeks	SP	NCT01445743
Halperin et al. [36]	2007–2011	RCT obsever-blind	Tdap/Td	135/138	Canada	≥30 weeks	SP	NCT00553228
Haidara et al. [24]	2019	RCT double-blind	Tdap/Td	133/67	Mali	14–27 weeks	GSK	NCT03589768
Hoang et al. [35]	2015	RCT	Tdap/TT	52/51	Vietnam	18–36 weeks	SP	NA

Note: RCT, randomized controlled trial; Tdap, tetanus–diphtheria–acellular pertussis vaccine; NA, not applicable; “--”, did not receive anything during pregnancy; GSK, GlaxoSmithKline; SP, Sanofi Pasteur; Td, tetanus–diphtheria vaccine; TT, tetanus toxoid vaccine.

**Table 2 vaccines-13-00666-t002:** Characteristics of included case-control studies.

Study ID	Study Period	Vaccine Efficacy (95% Confidence Interval)	N of Women (Exposure/Control)	Country	Gestational Age in Weeks of Vaccination	Major Vaccine Manufacturers
Merdrignac et al. [42]	2015–2019	87.0% [CI]: 55.0–96.0	75/201	Czech Republic, Ireland, Italy, Spain	Czech Republic: 28–36 weeks, Ireland: 16–36 weeks, Italy: >27 weeks, Spain: 27–36 weeks	NA
Fernandes et al. [39]	2015–2016	80.7% [CI]: 55.9–91.6	42/249	Brazil	Case group: 18–37 weeks Control group: 10–40 weeks	NA
Cheuvart et al. [26]	2011–2014	83.4% [CI]: 55.7–92.5	108/183	Australia	During pregnancy	GSK
Quinn et al. [8]	2012–2019	84.3% [CI]: 26.1–96.7	17/52	Australia	During pregnancy	NA
Godoy et al. [41]	2016–2018	88.0% [CI]: 53.8–96.5	47/124	Spain	During pregnancy	NA
Romanin et al. [40]	2012–2016	80.7% [CI]: 52.1–92.2	71/300	Argentina	During pregnancy	NA
López et al. [25]	2019–2021	81.0% [CI]: 14.0–96.0	50/150	Peru	27–36 weeks	NA
Dabrera et al. [38]	2012–2013	93.0% [CI]: 81.0–97.0	58/55	Britain	26–38 weeks	NA
Saul et al. [44]	2015–2016	69.0% [CI]: 13.0–89.0 *	117/117	Australia	28–32 weeks	GSK
Blasco et al. [43]	2015–2016	90.9% [CI]: 56.6–98.1 *	22/66	Spain	28–36 weeks	NA

Note: CI, confidence interval; GSK, GlaxoSmithKline; NA, not applicable; Case-control Study; “*” refers specifically to the vaccine effectiveness of infants under 3 months of age, and the rest is the vaccine effectiveness of infants under 2 months of age.

## Data Availability

All the data analyzed in this study are available in publicly available databases.

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
