# Peer review of "Efficacy, Immunogenicity, and Safety of Pertussis Vaccine During Pregnancy: A Meta-Analysis"

_vaccines, 2025, doi:10.3390/vaccines13070666_

Round 1

Reviewer 1 Report

Comments and Suggestions for Authors

Pertussis in the early childhood is still a challenging infectious disease that it can lead to hospitalization and lost parental working hours. In this manuscript, the authors conduct a meta-analysis regarding the efficacy of pertussis maternal vaccinations in the infants and the vaccination safety in mothers and in the neonatal population. The report is well conducted and clearly written. Herein are some comments for the authors:

  1. “Although substantial heterogeneity was observed in vaccine immunogenicity studies, subgroup analyses failed to identify specific sources of heterogeneity”. This phrase is not clear for the readers. Which are the subgroups of vaccine immunogenicity? Where is there heterogeneity?
  2. The authors have to report exactly the 3-components of GlaxoSmithKline vaccine and the 5-components of Sanofi Pasteur vaccines
  3. Please, specify the side effects that were controlled in that the studies that were included in metanalysis..
  4. The authors have to metanalyses if there is a predominant side effect in mothers and babies. The main question of the side effects in pregnant women if there was an increased risk of miscarriage and in neonates if there were any physical and mental problems in development
  5. The authors must comment the studies that were heterogenic. Which were the different conclusions between the heterogenic studies and the rest of the included reports?
  6. The authors have to report if there are any studies that provide clinical outcomes, eg reduction of pertussis infection in the first three months of life, less hospitalisations of the infected infants etc.

Author Response

Dear Reviewer,

We sincerely thank the editor and all reviewers for their valuable feedback that we have used to improve the quality of our manuscript. The reviewer comments are laid out below in italicized font and specific concerns have been numbered. Our response is given in red font.

Comments 1: “Although substantial heterogeneity was observed in vaccine immunogenicity studies, subgroup analyses failed to identify specific sources of heterogeneity”. This phrase is not clear for the readers. Which are the subgroups of vaccine immunogenicity? Where is there heterogeneity?

Response 1: Thank you for the feedback. We've revised the part about subgroup analyses on Page 7(lines 250–274), clearly stating the subgroups as vaccine manufacturers, vaccination gestational weeks, and control group interventions to address the heterogeneity issue. The following is the detailed content of the modification:

In the subgroup analysis stratified by vaccine manufacturers (GSK and SP), significant heterogeneity was observed in antibody responses across subgroups. For anti-PT antibodies in cord blood, the pooled standardized mean difference (SMD) for the GSK subgroup (4 studies) was 1.86 (95% CI: 1.56–2.17), significantly higher than the SP subgroup's SMD of 1.23 (95% CI: 0.89–1.57), with statistically significant between-group interaction (P = 0.036). In the analysis of anti-FHA antibodies in cord blood, the GSK subgroup (4 studies) exhibited a pooled SMD of 2.59 (95% CI: 2.27–2.91), which was significantly greater than the SP subgroup's SMD of 1.75 (95% CI: 0.98–2.51, P <0.001), with a trend toward statistical significance in interaction effect (P = 0.058). The intervention type of the control group had a limited impact on heterogeneity. For anti-PRN antibodies in cord blood, the placebo subgroup (5 studies) exhibited a pooled SMD of 2.18 (95% CI: 2.02–2.34) with no heterogeneity ( = 0%), whereas the TT/Td subgroup (3 studies) showed substantial heterogeneity (= 94.5%); however, the between-group interaction was not statistically significant (P = 0.952). In the analysis of anti-FHA antibodies, the placebo subgroup (4 studies) had a higher pooled SMD (2.49, 95% CI: 2.10–2.87) compared to the TT/Td subgroup (1.97, 95% CI: 1.12–2.82), though the interaction effect remained non-significant (P = 0.329). For vaccination gestational age (≥30 weeks and <30 weeks), a trend toward higher antibody levels was observed in the ≥30 weeks subgroup, though statistical significance was not reached. In the analysis of anti-FHA antibodies in cord blood, the pooled SMD for the ≥30 weeks subgroup (5 studies) was 2.40 (95% CI: 2.07–2.73), higher than the <30 weeks subgroup's SMD of 1.88 (95% CI: 0.28–3.49, 2 studies), with a non-significant interaction effect (P = 0.367). Similarly, for anti-PT antibodies, the ≥30 weeks subgroup (6 studies) exhibited a pooled SMD of 1.65 (95% CI: 1.24–2.05), greater than the <30 weeks subgroup's SMD of 1.37 (95% CI: 0.91–1.82, 2 studies), but the between-group interaction remained non-significant (P = 0.489).

Comments 2: The authors have to report exactly the 3-components of GlaxoSmithKline vaccine and the 5-components of Sanofi Pasteur vaccines.

Response 2: Thank you for pointing this out. We have explicitly specified in lines 347–351 of Page 11 that the GlaxoSmithKline vaccine contains three components (PT, PRN, FHA), and the Sanofi Pasteur vaccine includes five components (PT, PRN, FHA, DT, TT, with combined Fimbriae 2 and 3), as detailed in the revised manuscript.

Comments 3: Please, specify the side effects that were controlled in that the studies that were included in metanalysis.

Response 3: We have addressed the specification of monitored side effects in the included studies by supplementing the "2.4. Data Extraction" section (Page 3, lines 168–174). The revised text explicitly details serious adverse events (SAEs) tracked for both pregnant women and infants: 

In pregnant women, SAEs included maternal death, gestational hypertension, severe infections, preeclampsia, postpartum hemorrhage, and preterm birth. 

In infants, SAEs comprised neonatal death, congenital cardiac defects, respiratory distress syndrome, severe pneumonia, hyperbilirubinemia requiring phototherapy, febrile convulsions, dehydration, and hypoglycemia necessitating medical intervention. 

Comments 4: The authors have to metanalyses if there is a predominant side effect in mothers and babies. The main question of the side effects in pregnant women if there was an increased risk of miscarriage and in neonates if there were any physical and mental problems in development

Response 4: Thank you for bringing this to our attention. We acknowledge that the included studies had a limitation in follow-up duration, which regrettably prevented a comprehensive analysis of infant neurodevelopmental outcomes. For other serious adverse events relevant to the health of pregnant women and infants, detailed information can be found in the supplemented content on Page 3, lines 168-174:

Serious adverse events (SAEs) monitored in the pregnant women group included maternal death, gestational hypertension, severe infections, preeclampsia, postpartum hemorrhage, and preterm birth. For infants, SAEs comprised neonatal death, congenital cardiac defects, respiratory distress syndrome, severe pneumonia, hyperbilirubinemia requiring phototherapy, febrile convulsions, dehydration, and hypoglycemia necessitating medical intervention.

Comments 5: The authors must comment the studies that were heterogenic. Which were the different conclusions between the heterogenic studies and the rest of the included reports?

Response 5: Thank you for emphasizing the need to address study heterogeneity. We have expanded the discussion of heterogeneous studies in the revised manuscript, specifically on Pages 11–12 (lines 343–394), where we compare conclusions from divergent studies and explore potential causes of heterogeneity. The following is the detailed content of the modification:

Due to the observation of a large amount of heterogeneity in the study of vaccine immunogenicity, we conducted subgroup analysis. Taking umbilical cord blood anti-PT antibody as an example, the standardized mean difference (SMD) of the GSK vaccine group was 1.86 (95% CI: 1.56-2.17), which was significantly higher than that of the SP group (1.23, 95% CI: 0.89-1.57, P = 0.036). This finding may be associated with compositional differences between GlaxoSmithKline (3-components: PT, PRN, FHA) and Sanofi Pasteur (5-components: PT, PRN, FHA, DT and TT, Fimbriae 2 and 3 combined) vaccines, where tetanus and diphtheria toxoids in the formulations might influence pertussis component immunogenicity [21,45,46]. Furthermore, the antigen purity or adjuvant formula of GSK vaccines may be superior, thereby inducing a stronger IgG antibody response [34]. A similar trend was also observed in anti-FHA antibodies (standardized mean difference [SMD] = 2.59 in the GSK group vs. 1.75 in the SP group, P = 0.058). However, inter-manufacturer differences in anti-PRN antibodies were not statistically significant (P = 0.077), which may be attributed to the smaller number of included studies (only 4) or the more stable immunogenicity of PRN antigens. The impact of vaccine types used in the control group on heterogeneity differed across antibody types.  In the analysis of anti-PRN antibodies, heterogeneity in the placebo subgroup was extremely low (= 0%), whereas severe heterogeneity was observed in the TT/Td subgroup (= 94.5%). This may be attributed to potential cross-reactivity between tetanus toxoids in TT/Td vaccines and pertussis antigens, or substantial variations in production processes and vaccination doses of TT/Td vaccines across studies, leading to result fluctuations [47]. However, interaction tests showed no statistically significant intergroup differences (P = 0.952), indicating that the intervention type in the control group was not the primary source of heterogeneity.

After stratification by vaccination gestational age, antibody levels in the ≥30 weeks group were generally higher than those in the <30 weeks group, although heterogeneity remained significant. This may be associated with the higher efficiency of placental antibody transport during the third trimester of pregnancy [12]. However, the <30 weeks group had a small sample size (only 2 studies), limiting the stability of the results. Notably, even in the ≥30 weeks group, heterogeneity was not completely eliminated, suggesting that factors such as individual immune status and genetic background may influence vaccine response in addition to gestational age [48]. Differences in research design, antibody detection methods, and follow-up duration may all contribute to heterogeneity. For example, Perrett et al [33]. subdivided vaccination timing into 27–32 weeks and 33–36 weeks of gestation and found higher antibody levels in the latter subgroup, further supporting the impact of gestational age on immunogenicity. However, the inclusion of this study as two separate datasets in the analysis might have amplified statistical heterogeneity.

Comments 6: The authors have to report if there are any studies that provide clinical outcomes, eg reduction of pertussis infection in the first three months of life, less hospitalizations of the infected infants etc.

Response 6: Thank you for this important query. We have supplemented the discussion of clinical outcomes in the revised manuscript, specifically on Page 11 (lines 322–331), where we synthesize evidence from the studies we included. The following is the detailed content of the modification:

By synthesizing data from 10 case-control studies, the analysis revealed that maternal vaccination during pregnancy conferred an 85% vaccine effectiveness (VE, 95% CI: 78%–89%) in preventing pertussis infection in infants under 3 months of age, a vulnerable period before primary immunization is initiated. Notably, this protective effect translated directly into clinical outcomes: an Australian case-control study demonstrated a 94% reduction in hospitalization risk (95% CI: 59%–99%) among infected infants whose mothers received Tdap, highlighting the vaccine’s efficacy in preventing severe disease requiring medical intervention [44]. Concurrently, a UK study reported a 93% reduction in pertussis risk among newborns of vaccinated mothers, further linking maternal immunity to diminished severity of clinical presentations [38].

We hope that the revised version can address your concerns and demonstrate our sincere efforts to enhance the manuscript. Thank you again for your valuable time and constructive feedback.

Reviewer 2 Report

Comments and Suggestions for Authors

This paper reports a meta-analysis of 17 trials of pertussis vaccine during pregnancy. The outcomes considered are efficacy, immune responses, and safety. This all sounds good. The epidemiological and statistical methods are correct. The problem is that there is no need for this study, and it does not provide any new information to what is already known.

The key result of the meta-analysis is efficacy of 85% (78%, 89%) against pertussis in infants below 3 months of age following maternal immunization in late pregnancy. This is true. However, the target of pertussis vaccination of pregnant women is prevention of deaths and hospitalization due to pertussis in infants. The size of the present study material, 765 vaccine recipients and 682 controls, is much too small to examine those end points. Therefore, the whole exercise is next to useless.

For example, a systematic review from 2020 (Vygen-Bonnet S et al. BMC Infect Dis 2020;20:136) reviewed studies compromising 855,546 mother-infants’ pairs for efficacy. The result was 91%-94% vaccine effectiveness against hospitalization and 95% effectiveness for infant deaths. These are figures that matter and cannot be improved by re-analysis of old RCTs or case-control studies. Moreover, many of the cited studies in the present paper are really old, apart from being small size.

In summary, the result of the meta-analysis in the present study may be formally correct, but is useless. The meta-analysis of immune responses in different studies is futile to begin with and does not provide any meaningful information.

The meta-analyses for safety in mothers and infants is equally useless. From the clinical trials the only available end point is pooled SAEs, which it not informative at all. For example, the study of Vygen-Bonnet at al cited above reviewed 1.4 million vaccinated pregnant women for various clinical outcomes.

The present paper is written in odd language. It is correct English, but lacks thought. It appears that none of the authors has clinical or microbiological knowledge or perspective.

For example,

- The first sentence of Introduction mentions bacterium Pertussis. There is no such bacterium, but the correct microbiological name is Bordetella pertussis.

- The infection in infants < 3 months of age is not “a great strain on health care resources” as claimed by the authors in the 3rd sentence of Introduction, but the major argument for introduction of maternal immunization in prevention of infant deaths.

The last sentence of Introduction is total nonsense. This paper does not help to “optimize prenatal immunization strategies and enhance maternal-infant health outcomes”.

Altogether it seems that the paper may have with written by AI or ghost writer or both. There is no personal touch by the authors. My impression is that the present study is like a graduate seminar at a Chinese department of epidemiology. It is a methodological exercise, but not a real scientific contribution.

Author Response

Dear Reviewer,

We sincerely thank the editor and all reviewers for their valuable feedback that we have used to improve the quality of our manuscript. The reviewer comments are laid out below in italicized font and specific concerns have been numbered. Our response is given in red font.

Comments 1: The first sentence of Introduction mentions bacterium Pertussis. There is no such bacterium, but the correct microbiological name is Bordetella pertussis.

Response 1: Thank you for pointing out the nomenclature issue. We have corrected the first sentence of the Introduction to use the proper microbiological name "Bordetella pertussis" in line 77 of Page 2, as detailed in the revised manuscript.

Comments 2: The infection in infants < 3 months of age is not a great strain on health care resourcesas claimed by the authors in the 3rd sentence of Introduction, but the major argument for introduction of maternal immunization in prevention of infant deaths.

Response 2: Thank you for the feedback. We have revised the third sentence in the Introduction (Page 2, lines 80–82) to emphasize the mortality risk, now stating:

"Infection in this age group not only causes severe illness and family burden but also highlights the critical need for maternal immunization to prevent infant deaths." This better aligns with the primary rationale for maternal vaccination.

Comments 3: The last sentence of Introduction is total nonsense. This paper does not help to optimize prenatal immunization strategies and enhance maternal-infant health outcomes.

Response 3: Thank you for your critical feedback on the Introduction's concluding statement. We have revised the last sentence to better align with the study's objectives, now stating in Page 2 (lines 117–121) The following is the detailed content of the modification:

This study first evaluates the protective efficacy of maternal pertussis vaccination against infant infection, analyzing safety profiles, immunogenicity, and vaccine effectiveness in 3-month-old infants. By synthesizing these empirical findings, the research aims to provide a scientific basis for optimizing prenatal vaccination strategies and informing efforts to improve maternal-infant health outcomes.

We are truly grateful for your thorough and candid feedback on our manuscript. Your insights have been a wake - up call, helping us recognize the significant shortcomings in our work. We sincerely apologize for the deficiencies in the initial submission.

We have painstakingly incorporated your suggestions, as well as those from other reviewers, into the revised manuscript. Every correction was made with the utmost care to improve the quality, accuracy, and clarity of our work. We understand that our initial submission failed to meet the expected standards, but we are committed to learning from this experience. We hope that the revised version can address your concerns and demonstrate our sincere efforts to enhance the manuscript. Thank you again for your valuable time and constructive feedback.

Reviewer 3 Report

Comments and Suggestions for Authors

The paper is interesting and well written. The authors investigated the efficacy, immunogenicity, and safety of antenatal Tdap vaccination (tetanus, diphtheria, and acellular pertussis vaccine) in pregnant individuals through a meta-analysis confirming the need of the implementation of pertussis vaccination during pregnancy. The methodology, the statistical analysis, results and discussion are adequate and coerent with the endpoints. The paper may be acceptable for publication.

Author Response

Dear Reviewer,

We are deeply grateful for your positive feedback and constructive assessment of our manuscript. Your recognition of the study’s value, along with the confirmation that our methodology, analysis, and discussion align with the research objectives, is incredibly encouraging.  

Round 2

Reviewer 2 Report

Comments and Suggestions for Authors

Reject